# Topologically Protected Wormholes in Type-III Weyl Semimetal Co$_3$In$_2$X$_2$ (X = S, Se)

**Christopher Sims**

Department of Physics, University of Central Florida, Orlando, FL 32816, USA; christophersims@knights.ucf.edu

**Abstract:** The observation of wormholes has proven to be difficult in the field of astrophysics. However, with the discovery of novel topological quantum materials, it is possible to observe astrophysical and particle physics effects in condensed matter physics. It is proposed in this work that wormholes can exist in a type-III Weyl phase. In addition, these wormholes are topologically protected, making them feasible to create and measure in condensed matter systems. Finally, Co$_3$In$_2$X$_2$ (X = S, Se) are identified as ideal type-III Weyl semimetals and experiments are put forward to confirm the existence of a type-III Weyl phase.

**Keywords:** Weyl; wormhole; topological

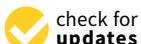



## 1. Introduction

The discovery of topological quantum materials has opened a large path in experimental condensed matter physics. Initially, the first material discovered was the topological insulator which has an insulating bulk and a topologically non trivial surface state which derived from the Dirac cone. Interestingly, these Dirac Fermions do not obey expected physics and behave relativistically [1–8]. Dirac fermions have later been seen to violate Lorentz symmetry and form tilted type-II and critically tilted type-III Dirac cones [9–11]. Tilted Dirac cones have been predicted to have the same physics as black holes in certain cases [12–15].

In the presence of additional perturbations, the formation of Weyl Fermions [16] and their surface Fermi arcs can be formed. These condensed matter excitations were first observed in condensed matter physics and have yet to be observed in high energy particle physics. Similarly to the Dirac cone, the edge states of the Weyl cone can be tilted to form type-II and type-III Weyl cones [17–20]. A plethora of type-I and type-II Weyl cones [21] have been discovered yet the discovery of a type-III Weyl Fermion phase has yet to be discovered conclusively [22,23].

Wormholes are a yet undiscovered but physically plausible object that can exist within the framework of general relativity [24,25]. Much work has been done in order to define wormhole topology, energy of formation, and experimental signatures. However, wormholes are largely considered improbable due to a need for a large amount of energy to form and maintain one within current models. The generation of a quasi-wormholes will prove valuable in understanding wormhole physics [26–28].

The type-III Dirac cone is predicted to host a direct analogue to a black hole where similar physics can be observed and measured with respect to the Dirac quasiparticles that experience the effects of the critically tilted Dirac cone. Limited experiments have been conducted in order to confirm the effects of the type-III Dirac phase and little to no materials have been discovered [29–34]. The counterpart to the Dirac black hole is the Weyl type wormhole phase [35–38]. In this work, it is predicted that wormholes can be formed and are topologically robust in the type-III Weyl phase. In addition, Co$_3$In$_2$S$_2$ and Co$_3$In$_2$Se$_2$ are predicted to host an ideal type-III Weyl fermion.

## 2. Materials and Methods

The band structure calculations were carried out using the density functional theory (DFT) program Quantum Espresso (QE) [39], with the generalized gradient approximation (GGA) [40] as the exchange correlation functional. Projector augmented wave (PAW) pseudo-potentials were generated utilizing PSlibrary [41]. The relaxed crystal structure was obtained from materials project [42,43] (for $Co_3In_2S_2$). The relaxed crystal for $Co_3In_2Se_2$ was calculated with QE. Crystal parameters that are calculated with DFT (Table 1) are compared to $Co_3In_2S_2$ (Table 2). The energy cutoff was set to 60 Ry (816 eV) and the charge density cutoff was set to 270 Ry (3673 eV) for the plane wave basis, with a k-mesh of $25 \times 25 \times 25$. High symmetry point K-path was generated with SSSP-SEEK path generator [44,45]. The bulk band structure was calculated from the "SCF" calculation by utilizing the "BANDS" flag in Quantum Espresso. As opposed to utilizing plotband.x included in the QE package, a custom python code is used to plot the band structure with the matplotlib package.

Single crystals of $Co_3In_2S_2$ and $Co_3In_2Se_2$ (SG: R$\bar{3}$M [166]) are grown via the Indium flux method [46,47]. Stoichiometric quantities of Co (99.9%, Alfa Aesar) and Se ($\sim$200 mesh, 99.9%, Alfa Aesar)/S ($\sim$325 mesh, 99.5%, Alfa Aesar) were mixed and ground together with a mortar and pestle. Indium (99.99% RotoMetals) was added in excess (50%) in order to allow for a flux growth. All precursor materials were sealed in a quartz tube under vacuum and placed inside a high temperature furnace. The sample was heated up to 1000 °C over 1440 min, kept at 1000 °C for 1440 min, cooled down to 950 °C over 180 min, kept at 950 °C for 2880 min, then slowly cooled down to 180 °C where the sample was taken out of the furnace then centrifuged. The grown crystals characterized via LEED (OCI LEED 600) (Figure 1B) and powder X-ray diffraction (XRD) (Bruker D8 DISCOVER, Cobalt Source) (Figure 2) to confirm their crystal structure (Figure 1A). The XRD results and calculations have been normalized; this results in some features in the calculated XRD being more prominent.

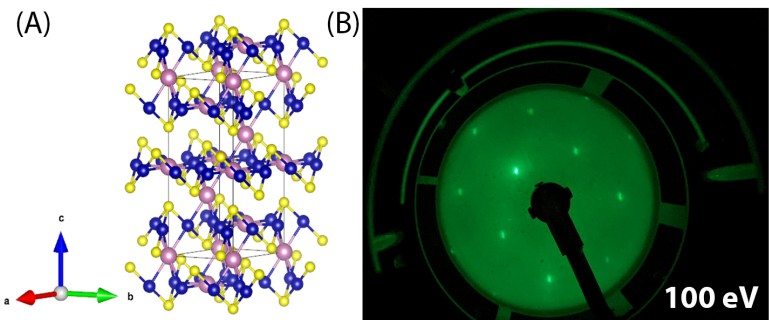

**Figure 1.** $Co_3In_2Se_2$ LEED: (**A**) Crystal Structure of $Co_3In_2X_2$ (X = S, Se) pink: In, blue: Co, yellow: Se, S (**B**) LEED image of cleaved $Co_3In_2Se_2$ showing hexagonal symmetry.

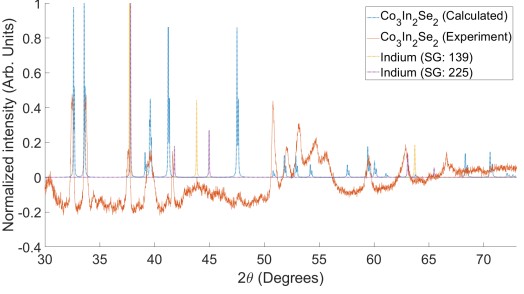

**Figure 2.** $Co_3In_2Se_2$ XRD: Calculated and experimental results of $Co_3In_2Se_2$. Both experiment and theory are normalized.

**Table 1.** Optimized cell parameters of $Co_3In_2Se_2$.

| Element | $Co_3In_2Se_2$ a (Crystal) | b (Crystal) | c (Crystal) |
|---|---|---|---|
| Se | 0.720007166 | 0.720007166 | 0.720007166 |
| Se | 0.279992834 | 0.279992834 | 0.279992834 |
| Co | 0.000000000 | 0.000000000 | 0.500000000 |
| Co | 0.500000000 | 0.000000000 | 0.000000000 |
| Co | 0.000000000 | 0.500000000 | 0.000000000 |
| In | 0.500000000 | 0.500000000 | 0.500000000 |
| In | 0.000000000 | 0.000000000 | 0.000000000 |
| | a (Å) | b (Å) | c (Å) |
| | 4.652649960 | 0.000029301 | 2.919783491 |
| | 1.614799853 | 4.363436112 | 2.919783491 |
| | 0.000042087 | 0.000029302 | 5.492930386 |

**Table 2.** Optimized cell parameters of $Co_3In_2S_2$.

| Element | $Co_3In_2S_2$ a (Crystal) | b (Crystal) | c (Crystal) |
|---|---|---|---|
| Se | 0.721000000 | 0.721000000 | 0.721000000 |
| Se | 0.279000000 | 0.279000000 | 0.279000000 |
| Co | 0.000000000 | 0.000000000 | 0.500000000 |
| Co | 0.500000000 | 0.000000000 | 0.000000000 |
| Co | 0.000000000 | 0.500000000 | 0.000000000 |
| In | 0.500000000 | 0.500000000 | 0.500000000 |
| In | 0.000000000 | 0.000000000 | 0.000000000 |
| | a (Å) | b (Å) | c (Å) |
| | 4.652816000 | 0.000000000 | 2.919838000 |
| | 1.614830000 | 4.363602000 | 2.91983732200 |
| | 0.000000000 | 0.000000000 | 5.493100000 |

## 3. Results

### 3.1. Realization of a Type-III Weyl Phase

The electronic nature of the Weyl cone can be visualized by the Landau level dispersion. By constructing a simple Hamiltonian (see Supplementary Information for details) it is possible to model the tilting of the Weyl cone. For the condition of a type-I Weyl where $C < |1|$ we select C = 0.5 (Figure 3A). Here we see that the Weyl cone is slightly tilted but is still preserves lorentz invariance. $C = 1$ (Figure 3B) shows a similar dispersion. When $C = 5$ (Figure 3C), the Weyl cone breaks lorentz invaraince and forms a type-II over tilted Weyl dispersion. When $C = -1$ The Weyl cone becomes critically tilted (Figure 3D) and forms a type-III Weyl cone. In the type-III Weyl phase it can be seen that the chiral edge mode has a linear dispersion in $k$-space and transitions from the hole-band to the electron-band.

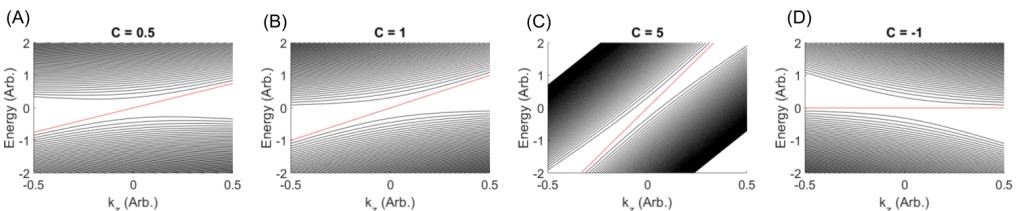

**Figure 3.** Landau levels: (**A**) $C = 0.5$ Type-I Weyl. (**B**) $C = 1$ Type-I Weyl. (**C**) $C = 5$ Type-II Weyl Semimetal. (**D**) $C = -1$ Type-III Weyl Semimetal.

### 3.2. Wormhole Experimental Signatures

#### 3.2.1. ARPES

Angle resolved photoemission spectroscopy (ARPES) is a valuable method to probe the Weyl states in order to discover a type-III weyl phase in predicted materials. The ideal ARPES signature of a type-III Weyl phase is a Weyl line that connects a hole-like conduction band to an electron like valance band (Table 3). In addition, this line must not be parabolic for a certain dispersion in $k$ (crystal symmetry can preserve this condition). The Fermi surface of a type-III Dirac and a type-III Weyl state will give similar features that cannot be distinguished without spin-resolved ARPES (an enclosed loop or line between two points). Thus, in order to confirm the topological nature of the Weyl line node it would be ideal to conduct spin-resolved measurement to probe the spin states along the linear Weyl line. Weyl arcs only posses one spin per arc; however, Dirac cones possess two spins per arc that flip at the Dirac point. In the critically tilted type-III Dirac and Weyl phases this will resolve into only one spin per arc in the Weyl phase.

**Table 3.** Features of different Weyl cones.

|  | **Type-I** | **Type-II** | **Type-III** |
|---|---|---|---|
| Dispersion | Weyl Cones | Overtilted Weyl cones | Critically tilted Weyl cones |
| Fermi surface ($K_x$,$K_y$) | Fermi arc | Fermi arc | Weyl line |
| DOS ($E_F$) | singularity | $e$ and $h$ pocket | Weyl line |
| Fermi arc | yes | yes | yes |
| Wormhole analogue | Open WH | Pinched WH | Traversable WH |
| Typical Materials | TaAs, NbAs | WTe$_2$ | Co$_3$In$_2$Se$_2$, Co$_3$In$_2$S$_2$ |

#### 3.2.2. Wormhole Anomaly in Magnetoresistance

Another way to confirm the existence of topological wormholes is to perform magnetoresistance measurements in type-III Weyl semimetals that are nonmagnetic. When measuring the longitudinal resistance of a material as the magnetic field is rotated, it is possible to measure the anisotropy in the sample in order to gain insight to the magnetoresponse in relation to different crystal axis. This response is typically called butterfly magnetoresistance because of how the anisotropy typically looks when plotted on a polar plot [48–50]. The magnetoresistance is a function of the electron (hole) mobility in the sample. The mobility is also correlated with the Fermi velocity. We know from previous work that electrons (holes) that exist in the flat band will have zero Fermi velocity; this will lead to no mangetoresponse at the angle where the DOS of the type-III Weyl cone lines up with the magnetic field (Figure 4A). In order to simulate this response we construct a toy model of the variable Fermi velocity as a function of angle for different chemical potentials by using trigonometric functions. The actual magnetoresisistance can be measured by magnetotransport. In a type-III Weyl semimetal which is composed of two pairs of Weyl points, we expect to see typical butterfly magnetoresponse at the Weyl line, but as the chemical potential moves away from the Weyl line level we expect to see that response to decrease (Figure 4B,C) and show less of an intense mangetoresponse. The Fermi level can be adjusted by backgating, top gating, or a combination of the two in order to access the Weyl line states and to measure the electrical respose (e.g., R$_{xx}$ vs. V$_{topgate}$, R$_{xx}$ vs. V$_{backgate}$). In the case of magnetic type-III WSMs (Co$_3$In$_2$Se$_2$, Co$_3$In$_2$S$_2$), the Weyl line contribution can be convoluted with the normal magnetic response of the material. Type-III Weyl states exist above the Fermi level in both Co$_3$In$_2$Se$_2$ and Co$_3$In$_2$S$_2$. In order for these states to be more accessible, the chemical potential can be tuned by methods such as potassium doping K$_x$Co$_3$In$_2$Se$_2$ (X $\leq$ 0.1) or impurity doping Co$_3$In$_2$Se$_{2-x}$I$_x$ (X $\leq$ 0.1).

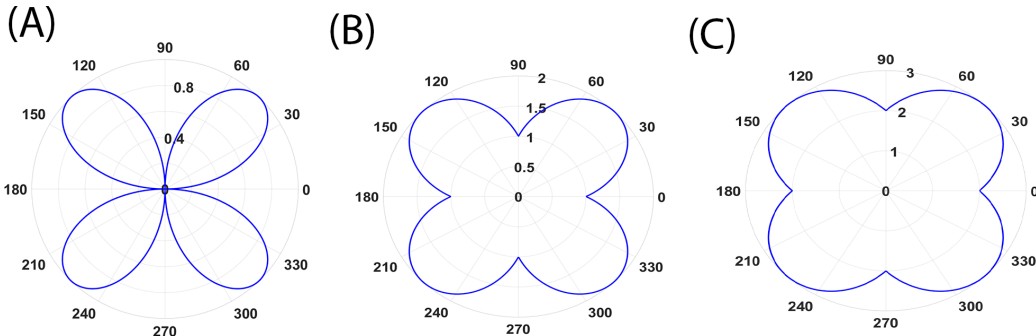

**Figure 4.** Butterfly magnetoresistance: (**A**) Magnetoresistance near the Weyl line energy level E (**B**) E + ΔE a small energy difference from the Weyl line (**C**) E + ΔE a large energy difference from the Weyl line level.

## 4. Discussion

In order to form a type-III Weyl phase in a crystal lattice, it is necessary to satisfy several conditions. Firstly, perfectly flat bands must exist with a large enough momentum dispersion to connect two bands (or a band must be flat for a period between these two bands). The band must be chiral and connect a hole-like band to an electron-like band; this condition allows for inversion symmetry to be preserved (from band inversion). Materials that satisfy this condition only satisfy a type-III Dirac semimetal phase; therefore, in order to break time reversal symmetry and turn the Dirac cone into two Weyl nodes, magnetism (or spin orbit interactions) must also exist in these materials in order for there to be a type-III Weyl phase. The best materials that satisfy these conditions are Kagome magnets. Kagome materials are known for having flat bands near the Fermi level which upon the inclusion of magnetism turn into Weyl semimetals. Recently, $Co_3Sn_2S_2$ has gained a great interest for being a Weyl type Kagome material that hosts a Weyl line. However, this Weyl line has been determined to be composed of a nearly critically tilted type-I Weyl cone [51–55]. In the case of $Co_3Sn_2S_2$ type Kagome systems, the R$\bar{3}$m (No. 166) space group can protect the existence of flat bands. This work identifies $Co_3In_2S_2$ (Table 2) and $Co_3In_2Se_2$ (Table 1) as excellent candidates that host flat bands near the Fermi level. $Co_3In_2S_2$ has perfectly flat type-III Weyl cones at both ∼150 meV and ∼350 meV above the Fermi level upon the inclusion of magnetism (Figure 5A,B).

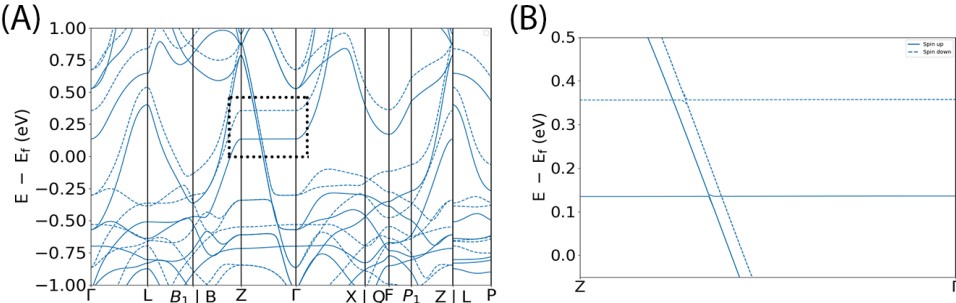

**Figure 5.** $Co_3In_2S_2$ band structure: (**A**) Bulk band structure of $Co_3In_2S_2$ (**B**) Zoomed-in view of the Z-Γ high symmetry line.

It is well understood that Dirac Fermions can be described as a superposition of two Weyl Fermions of opposite chirality [16] with short range entanglement. There have been several ongoing experiments in order to discover the existence of a quasi-black hole in the type-III Dirac phase [29,34,56]. If a black hole excitation is found in a type-III Dirac semimetal and a wormhole is discovered in a type-III Weyl semimetal, this supports the Einstein–Rosen (ER)= Einstein–Podolsky–Rosen (EPR) conjecture in astrophysics [57,58]

within a condensed matter system. The ER=EPR conjecture states that wormholes are two entangled black holes.

## 5. Conclusions

In conclusion, this work has outlined the parameters that can allow for a topologically protected wormhole in a type-III Weyl semimetal. Several experimental measurements are put forward as a test of a type-III Weyl phase, such as ARPES and electronic transport. Finally, materials $Co_3In_2S_2$ and $Co_3In_2Se_2$ are identified as materials that host flat bands needed for critically tilted type-III Weyl cones. $Co_3In_2Se_2$ is discovered as a new magnetic Kagome material via crystal synthesis and DFT prediction. Finally, it is postulated that the ER=EPR conjecture can be confirmed by discovering a black hole excitation in a type-III Dirac semimetal and a wormhole excitation in a type-III Weyl semimetal. This future research direction may provide the insight needed in order to unify general relativity and quantum mechanics.

**Supplementary Materials:** The following are available online at https://www.mdpi.com/article/10.3390/condmat6020018/s1, Description of Weyl points and a derivation of Landau Levels.

**Funding:** This research received no external funding.

**Data Availability Statement:** Data used in this text is available upon request.

**Acknowledgments:** The author acknowledges the University of Central Florida Advanced Research Computing Center (ARCC) for providing computational resources and support that have contributed to results reported herein. This work acknowledges Lawrence Berkeley National Laboratory (LBNL) Molecular Foundry for the characterization of grown crystals. Correspondence should be addressed to C.S. (Email: ChristopherSims@knights.ucf.edu).

**Conflicts of Interest:** The author declares no conflict of interest.

**Sample Availability:** Samples of compounds $Co_3In_2S_2$ and $Co_3In_2Se_2$ are available upon reasonable request.

## Abbreviations

The following abbreviations are used in this manuscript:

| | |
|---|---|
| WSM | Weyl Semimetal |
| DSM | Dirac semimetal |
| DFT | Density Functional Theory |
| WH | Wormhole |
| DOS | Density of states |

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
