# Peer review of "Topologically Protected Wormholes in Type-III Weyl Semimetal Co3In2X2 (X = S, Se)"

_condensedmatter, doi:10.3390/condmat6020018_

Round 1

Reviewer 1 Report

Dear Authors, I personally liked the paper.

I have only a minor suggestion: in my personal opinion it would be fantastic a brief comment on the ER=EPR conjecture by Susskind here and/or for future works, if it is feasible with wormholes in a type-III Weyl phase. Of course this is not crucial for the acceptance of your work, from my personal point of view. 
This would be - maybe in the future - a point where to focus as it is one of the most hottest topics for quantum gravity. There is a lot of stuff down there.

With Best Regards

Reviewer 2 Report

In this manuscript the author presents the growth of Co3In2S2, Co3In2Se2 single crystals and propose that they are type-III Weyl semimetals with ideal flat bands.

The content of the research is to some extent interesting, but the style to present the results really need to be significantly improved. In the current version of manuscript, the lengthy discussions of "wormhole physics" (which are not original) takes more than 90% of the main text whereas the original contents, i.e., the DFT calculations and the growth/characterization of Co3In2S2, Co3In2Se2, are not discussed in detail. 

I recommend that the author rewrite the manuscript completely, focusing on the original work and presenting them clearly with details. I would like to reconsider it only after such a major revision.

Other comments:
- Equation (5) needs to be explained clearly.
- Equation (6) is not correct (try to invert the conductivity tensor carefully in view of \sigma_{xy}=-\sigma_{yx}) 

Reviewer 3 Report

Topologically Protected Wormholes in Type-III Weyl Semimetal Co3In2X2 (X = S, Se)

by Christopher Sims

The author of this paper studied the gap structure in Co3In2X2 (X = S, Se)
using the commercialized packet Quantum Espresso for a DFT calculation and the Angle resolved photoemission spectroscopy (ARPES) for experimental observation of the type-III Weyl semimetal state.

The author showed that by an appropriate choice of a Hamiltonian (Eq. 4) for the DFT input, types I, II and III of Weyl states can be obtained depending on the value of C. These are shown in Fig. 1. My remark on this part is:
The author mentioned the references 42 and 43 for these calculations. He should make clear what was taken from these works and what is the contribution of the present work. Also, I guess that the Hamiltonian (4) is a known Hamiltonian which generates the Dirac cones. If this is true, what is the contribution of the author to obtain the figure 1?  The description of the DFT packages and parameters used in the Quantum Espresso progrram in section 2 is so short and not physically explained.

The experimental technique was described in section 2. The grown crystals  of Co3In2S2, Co3In2Se2, Ni3In2S2, and Ni3In2Se2 were characterized via LEED [Fig A2(B)] and powder XRD [Fig A1] to confirm their crystal  structure.  The author said in sect. 3.2.1 that the use of ``Angle resolved photoemission spectroscopy (ARPES)'' allows to discover the Weyl  phase in  Co3In2X2 (X = S, Se).  But then the author wrote ``In order to confirm the topological nature of the Weyl line node it would be ideal to conduct spin-resolved measurement to probe the spin states along the linearWeyl line. This measurement can take place at a synchrotron source with a Mott spin detector or with a pump-probe laser based setup utilizing a circularly polarized pump in order to preferentially select spins to detect with the probe laser. Since the Fermi surface of a type-III Dirac and a type-III Weyl state will look the same without spin-resolved ARPES (an enclosed loop or line between two
points) it is necessary to conduct this measurement. With spin-resolved ARPES, a phase
 shift of p (1 spin per arc) cross the entire loop will indicate a type-III Weyl phase. A
 type-III Dirac cone will have a 2p (2 spin states [up,down] per arc) spin phase shift.''.
 This paragraph is confusing: all of these (spin-resolved measurements) are done or will be done?

In addition, how did the author obtain the figures 2?  Solving Eqs. 5&6 needs several hypotheses on g and v which are not specified.
At the end of section 3 the author used the future tense ``In the case of magnetic type-III
WSMs (Co3In2Se2, Co3In2S2), this feature will be convoluted with the normal magnetic
response of the material.''  Again, I do not understand if these have been checked or will be checked.

As for Fig. 3, the author did not say how he obtained it and with which parameters.

In conclusion, while the problem treated here has some interest, the presentation of the results lacks the clarity. I do not recommend the publication of this paper in its present form.

Round 2

Reviewer 2 Report

I would like to thank the author for addressing my comments in the revised manuscript. I recommend publishing the current version after minor revision in response to the comments bellow:

- In Section 3.2.2, the author proposes to take the angular dependence of magnetoresistance as an evidence to confirm the presence of flat Weyl line. However, according to the DFT results in Fig. 5(a), the Weyl cones of Co3In2S2 are actually located above the Fermi level, which means that they are not occupied and would not contribute to the magnetoresistance. Please address this issue in the manuscript.

- The font size of the labels in Fig. 2 and Fig. 4 are way too small. Please enlarge the font size so that they will be clearly visible in the printed version. 

Reviewer 3 Report

The author has included new elements in replying satisfactorily to my previous remarks. There are however some problems of references at the end of section 4. After corrections, I recommend to publish this paper in Condensed Matter.
